# Activation of Nrf2/HO-1 Antioxidant Pathway by Heme Attenuates Calcification of Human Lens Epithelial Cells

**DOI:** 10.3390/ph15050493

**Published:** 2022-04-19

**Authors:** Arpan Chowdhury, Enikő Balogh, Haneen Ababneh, Andrea Tóth, Viktória Jeney

**Affiliations:** 1MTA-DE Lendület Vascular Pathophysiology Research Group, Research Centre for Molecular Medicine, Faculty of Medicine, University of Debrecen, 4032 Debrecen, Hungary; chowdhury.arpan@med.unideb.hu (A.C.); balogh.eniko@med.unideb.hu (E.B.); hannen.ababneh@med.unideb.hu (H.A.); andrea.toth@med.unideb.hu (A.T.); 2Doctoral School of Molecular Cell and Immune Biology, Faculty of Medicine, University of Debrecen, 4032 Debrecen, Hungary

**Keywords:** cataract, lens epithelial cells, osteogenic differentiation, lens calcification, nuclear factor erythroid 2-related factor 2 (Nrf2), heme, heme oxygenase-1 (HO-1), iron

## Abstract

Cataract, an opacification in the crystalline lens, is a leading cause of blindness. Deposition of hydroxyapatite occurs in a cataractous lens that could be the consequence of osteogenic differentiation of lens epithelial cells (LECs). Nuclear factor erythroid 2-related factor 2 (Nrf2) controls the transcription of a wide range of cytoprotective genes. Nrf2 upregulation attenuates cataract formation. Here we aimed to investigate the effect of Nrf2 system upregulation in LECs calcification. We induced osteogenic differentiation of human LECs (HuLECs) with increased phosphate and calcium-containing osteogenic medium (OM). OM-induced calcium and osteocalcin deposition in HuLECs. We used heme to activate Nrf2, which strongly upregulated the expression of Nrf2 and heme oxygenase-1 (HO-1). Heme-mediated Nrf2 activation was dependent on the production of reactive oxygens species. Heme inhibited Ca deposition, and the OM-induced increase of osteogenic markers, RUNX2, alkaline phosphatase, and OCN. Anti-calcification effect of heme was lost when the transcriptional activity of Nrf2 or the enzyme activity of HO-1 was blocked with pharmacological inhibitors. Among products of HO-1 catalyzed heme degradation iron mimicked the anti-calcification effect of heme. We concluded that heme-induced upregulation of the Nrf2/HO-1 system inhibits HuLECs calcification through the liberation of heme iron.

## 1. Introduction

Cataract, an opacification in the lens of the eye, is a leading cause of blindness, mostly in developing countries [1]. Cataracts are an aging disease, supported by the fact that more than half of American people age 80 or older either have cataracts or have had cataract surgery according to the National Eye Institute. Cataracts are a huge global health burden that is estimated to further increase because of population growth and aging. Currently, there is no pharmacological intervention to prevent or cure cataracts, the only treatment option is the surgical removal of the cataractous lens.

The ocular lens is an avascular and transparent tissue with very high protein content [2]. A single layer of cuboidal-shaped lens epithelial cells (LECs) covers the anterior lens surface. These parental cells follow a specific developmental pattern and are responsible for the growth and development of the entire lens [2]. LECs differentiate into long, thin, and transparent fiber cells, which lack a nucleus and other cellular organelles. Fiber cells are tightly packed and compose the bulk of the lens [2]. On the other hand, tissue injury or certain stimuli such as transforming growth factor beta can trigger epithelial-to-mesenchymal transition (EMT) of LECs [3]. Growing evidence suggest the role of EMT and fibrosis in the opacification of intraocular lenses and in the formation of anterior subcapsular cataract [3,4].

Surprisingly, some studies identified calcium and phosphate containing hydroxyapatite, the major inorganic component of bone tissue, in senile cataractous lenses [5,6,7]. Deposition of hydroxyapatite in soft tissues or so-called ectopic calcification is a pathologic event that can occur in both genetic and acquired clinical conditions. For a long time, ectopic calcification was considered to be a passive process but recent discoveries proved that it is not the case, and identified numerous regulators of the calcification process. For example, studies on patients with rare inherited calcification syndromes such as pseudoxanthoma elasticum, generalized arterial calcification in infancy, and arterial calcification due to deficiency of CD73 led to the identification of inorganic pyrophosphate (PPi) as the principal physiologic inhibitor of hydroxyapatite deposition in soft tissues [8,9,10,11]. These hereditary disorders are caused by monogenic mutations in genes encoding key proteins in PPi metabolism such as ATP binding cassette subfamily C member 6, ectonucleotide pyrophosphatase/phosphodiesterase 1, and CD73 proteins, leading to reduced circulating levels of PPi [8,9,10,11].

Besides the genetic origin, ectopic calcification, especially vascular calcification is associated with chronic diseases such as renal failure and diabetes [12]. Additionally, aging is also an important predisposing factor for vascular calcification [13]. Vascular calcification is a cell-regulated process driven by an osteochondrogenic trans-differentiation of vascular smooth muscle cells (VSMCs) [14]. The osteochondrogenic trans-differentiation of VSMCs is orchestrated by the master transcription factor of osteogenesis, known as runt-related transcription factor 2 (Runx2) [15].

Previously we showed that LECs, similarly to VSMCs, are able to undergo osteochondrogenic trans-differentiation characterized by Runx2 upregulation and extracellular matrix mineralization when exposed to high inorganic phosphate and calcium. Moreover, we showed the presence of osteocalcin (OCN), a major non-collagenous bone matrix component in cataractous lenses, suggesting that osteochondrogenic trans-differentiation of LECs may play a role in lens calcification [16].

Unbalance between the formation and elimination of reactive oxygen species (ROS) is largely implicated in the pathomechanism of age-related diseases including cataract formation and vascular calcification [17,18]. Nuclear factor erythroid-2-related factor 2 (Nrf2) is a central transcription factor in the regulation of redox homeostasis and cytoprotection. Under stress-free conditions, Nrf2 binds to its negative regulator, Kelch-like ECH-associated protein 1 (Keap1), and undergoes polyubiquitinylation and proteasomal degradation [19]. Under stress, the Nrf2-Keap1 interaction is disrupted, leading to Nrf2 stabilization, nuclear translocation, and binding of Nrf2 to the antioxidant response elements of hundreds of Nrf2-target genes with diverse biological functions [19].

Heme oxygenase-1 (HO-1) is an Nrf2-regulated cytoprotective molecule with particular importance [20]. HO-1 catalyzes heme degradation into carbon monoxide (CO), ferrous ion (Fe^2+^), and biliverdin which latter is promptly converted to bilirubin. Cytoprotective actions of HO-1 are mostly based on the ability of HO-1 to eliminate heme and the antioxidant and anti-inflammatory properties of the above-mentioned heme degradation products [20]. Loss of Nrf2-dependent cytoprotection is implicated in the formation of age-related cataracts [21]. Moreover, overexpression of a negative dominant mutant of HO-1 lacking HO-1 enzyme activity induces early-onset nuclear cataracts [22]. On the contrary, upregulation of the Nrf2/HO-1 axis prevents cataract development and progression in diabetic rats [23]. Previous reports showed that upregulation of the Nrf2/HO-1 system attenuates osteochondrogenic trans-differentiation and calcification of VSMCs and valve interstitial cells [24,25,26,27,28]. Here we aimed to explore whether the induction of the Nrf2/HO-1 antioxidant pathway influences osteochondrogenic differentiation and extracellular matrix (ECM) mineralization of LECs.

## 2. Results

### 2.1. Osteogenic Stimuli Induces ECM Calcification of HuLECs

Previously we showed that elevated inorganic phosphate and calcium induce osteochondrogenic differentiation and ECM calcification of HuLECs in a synergistic manner, and proposed that this mechanism might contribute to hydroxyapatite formation in the cataractous lens [16]. Here, we implemented an in vitro model of lens calcification in which we cultured HuLECs under control (Ctrl) and osteogenic (OM; 2.5 mmol/L Pi, 0.3 mmol/L Ca) conditions for up to six days. ECM calcification started on day3 and kept on increasing till day6 in OM-treated HuLECs, whereas we did not detect ECM calcification in HuLECs under Ctrl conditions (Figure 1a,b). To confirm the result of AR staining we measured calcium levels from HCl-solubilized ECM samples, which revealed that ECM calcium content was more than 16-fold higher in OM-treated HuLECs than the controls (Figure 1c). Finally, we measured the level of OCN, the bone-specific calcium-binding protein, in EDTA-solubilized ECM of HuLECs. In response to the osteogenic stimuli, OCN levels increased by 8-fold compared to control (Figure 1d).

### 2.2. Heme Induces the Nrf2/HO-1 Axis in HuLECs in a ROS-Dependent Manner

Previously it has been shown that activation of the Nrf2/HO-1 axis prevents high Pi-induced calcification of vascular smooth muscle cells and valve interstitial cells [27,28]. Based on these observations here we aimed to test whether the activation of the Nrf2/HO-1 system in HuLECs inhibits HuLECs calcification. Based on previous publications we used heme at a supraphysiological concentration of 1–50 µmol/L to activate the Nrf2/HO-1 antioxidant pathway. We showed that heme dose-dependently upregulated Nrf2 expression in HuLECs (Figure 2a,b). We investigated the heme-induced expression of two Nrf2 target genes, NQO1 and HO-1. We found a modest (~2-fold) NQO1 upregulation and a robust increase of HO-1 (~200-fold) by the highest concentration of heme (50 µmol/L) (Figure 2c,d).

In many cell types, heme serves as a pro-oxidant through the participation of the heme iron in Fenton reaction-driven ROS production. There is a complex interplay between Nrf2 and oxidative stress in which ROS activates Nrf2 and Nrf2 target genes control oxidative stress. Based on these facts, next we investigated the ROS-dependency of heme-induced Nrf2 upregulation in HuLECs. First, we showed that heme induces ROS production in HuLECs (Figure 3a). The glutathione precursor N-acetylcysteine (NAC) completely abolished heme-induced ROS formation in HuLECs (Figure 3b). Moreover, NAC attenuated heme-mediated upregulation of Nrf2 in HuLECs (Figure 3c,d).

### 2.3. Heme Inhibits Osteogenic Stimuli-Induced ECM Calcification of HuLECs

Heme has been shown to inhibit osteochondrogenic differentiation and calcification of vascular smooth muscle cells as well as valve interstitial cells [27,28]. Therefore next we investigated whether heme attenuates OM-induced calcification of HuLECs. As revealed by AR staining and determination of calcium in HCl-solubilized ECM, heme decreased OM-induced ECM calcification of HuLECs in a dose-dependent manner (Figure 4a–c). Then we investigated the effect of heme on crucial markers of the osteogenic trans-differentiation process. Heme (25 µmol/L) not only abolished the OM-induced increase of RUNX2, the master transcription factor of osteogenesis, but lowered its expression below the control level (Figure 4d,e). Heme (25 µmol/L) downregulated the expression of ALP compared to control and OM-induced HuLECs (Figure 4d,f). Osteogenic stimuli induced a robust increase in the expression of OCN in the ECM of HuLECs, which was completely abolished by heme (Figure 4g).

### 2.4. The Nrf2/HO-1 Antioxidant Pathway Plays an Essential Role in Heme-Mediated Calcification Inhibition

In the following experiments, we addressed whether the induction of the Nrf2/HO-1 pathway is involved in the heme-mediated inhibition of HuLECs’ calcification. We used heme (10 µmol/L) to induce, and ML385 (10 µmol/L) to inhibit Nrf2, and investigated the effect of Nrf2 modulation on HuLEC calcification. Calcification of ML385-treated HuLECs was stronger than vehicle-treated cells (Figure 5a,b). Additionally, in the presence of ML385 heme was unable to prevent OM-induced calcification of HuLECs (Figure 5a,b). These results suggest that Nrf2 plays a protective role in the calcification process in HuLECs and that upregulation of Nrf2 by heme establishes the anti-calcification effect of heme (Figure 5a,b).

Previously we showed that HO-1 enzyme activity is required for heme-mediated attenuation of calcification in valve interstitial cells [27]. So, next, we addressed the question of whether HO-1 is involved in the anti-calcification effect of heme in HuLECs. We used heme to induce, SnPPIX and ZnPPIX to inhibit HO-1 activity. In the absence of HO-1 inhibitors, heme (10 µmol/L) abolished OM-induced calcification of HuLECs as assessed by AR staining and calcium measurement (Figure 6a,b). On the contrary, heme-mediated calcification inhibition was lost in the presence of HO-1 inhibitors (Figure 6a,b). These results suggest that an intact Nrf2/HO-1 system and HO-1 enzyme activity are needed for the anti-calcification effect of heme.

### 2.5. Heme Degradation Products Possess Anti-Calcification Activities

Abolishing HO-1 activity ruins the inhibitory effect of heme on calcification, suggesting that heme degradation products may play a role in heme-induced inhibition of calcification. Heme degradation by HO-1 has three products, iron (Fe), CO, and biliverdin—which is quickly converted to bilirubin (BR) by biliverdin reductase. Therefore, in our next experiment, we investigated the effect of these heme degradation products on HuLEC calcification. We treated HuLECs with OM in the presence or absence of BR (5–50 µmol/L), Fe (5–50 µmol/L), and CO which was administered as CORM2 (25–200 µmol/L). AR staining and ECM calcium measurements revealed that iron is the ultimate heme degradation product that exhibits a strong anti-calcification potential (Figure 7). BR showed limited and CORM showed no anti-calcification potential in OM-treated HuLECs (Figure 7).

## 3. Discussion

Extracellular matrix calcification, a tightly regulated process, is an intrinsic feature of osteoblasts and chondrocytes and plays an essential role in the growth, repair, and remodeling of bones and cartilage. On the other hand, bone-like extracellular matrix deposition can be observed in extra-skeletal soft tissues which has been assumed to be a passive process for a long time. In the past two decades, this paradigm was shifted and nowadays ectopic calcifications, particularly vascular and valve calcifications, are considered cell-mediated, highly regulated processes sharing many aspects with physiological bone formation [29].

Investigation of cataractous lenses with Fourier transform infrared and Raman mic-rospectroscopies revealed the presence of hydroxiapatite in senile cataractous lenses [5,6,7], although the current knowledge concerning cataract formation did not provide an explanation for this phenomenon. In our previous study, we showed that HuLECs can undergo osteochondrogenic differentiation and ECM calcification upon osteogenic stimulation [16]. We detected high calcium levels and OCN, the major non-collagenous bone protein in human cataractous lenses which supports our hypothesis that calcification of HuLECs might occur in vivo and that this active mechanism might be implicated in the formation of hydroxyapatite in cataractous lenses [16].

Oxidative stress plays a critical role in the development of age-related eye diseases including cataract formation. The Keap1/Nrf2/ARE system is a cellular defense mechanism against oxidative stress with enormous importance. Studies highlighted the protective role of Nrf2-driven cytoprotective mechanisms in different age-related eye diseases including glaucoma, macular degeneration, diabetic retinopathy, and cataract [30,31]. Gao et al. investigated the protein and gene expression of Nrf2 and Keap1 in human lenses. They found that the expression of Nrf2 was significantly lower, and Keap1 was higher in lenses derived from elderly people (65–80-year old) compared to a younger population [32]. This expression pattern leads to deterioration of oxidative stress defense in the aging eye which can contribute to the development of age-related eye diseases such as cataracts. Accumulating evidence suggest that Nrf2 is a promising therapeutic target for the prevention of age-related cataracts [31,33].

HO-1 is the inducible, rate-limiting enzyme of heme degradation with diverse antioxidant and anti-inflammatory capabilities [20]. Regarding lens epithelial cells, HO-1 has been shown to protect HuLECs from hydrogen-peroxide-induced oxidative stress and apoptosis [34]. Loss of function mutation of the HO-1 gene induces early-onset nuclear cataracts through the activation of oxidative and endoplasmic reticulum stress [22].

Here we investigated the effect of the activation of the Nrf2/HO-1 pathway on the osteogenic differentiation of HuLECs. We used heme, the ubiquitous iron compound to upregulate the Nrf2/HO-1 axis (Figure 2). Iron compounds may facilitate hydroxyl-radical generation from activated oxygen species, such as hydrogen peroxide in the Fenton reaction [35]. Earlier work showed that heme is a biological Fenton reagent that can participate in unfettered ROS production [36]. ROS plays a critical role in the regulation of the Keap1/Nrf2 system. Keap1 is a thiol-rich protein that binds Nrf2 and promotes its degradation under unstressed conditions [37]. Excess ROS directly modifies the conformation of Keap1, allowing Nrf2 stabilization and nuclear translocation [37].

Here we showed that the ROS scavenger NAC decreased heme-induced upregulation of Nrf2 expression as expected (Figure 3). Heme dose-dependently inhibited OM-induced ECM calcification of HuLECs (Figure 4). Moreover, heme inhibited OM-induced upregulation of RUNX2, the master transcription factor of osteogenesis, and downregulated the expression of ALP in HuLECs (Figure 4). These results are in agreement with previous reports on the inhibitory effect of heme on phosphate-induced calcification of VSMCs and VICs [27,28].

To understand the role of Nrf2 in the anti-calcification effect of heme we used ML385, an Nrf2 inhibitor that interacts with Nrf2 and prevents the DNA binding activity of the Nrf2-containing heterodimeric transcription factors. In the presence of ML385 heme lost its ability to inhibit calcification of HuLECs (Figure 5).

Activation of the Nrf2 system by different agents such as dimethyl fumarate, hydrogen sulfide, tert-butylhydroquinone, rosmarinic acid, or mitoquinone has been shown to decrease VSMCs calcification [24,38,39,40,41]. Among them, dimethyl fumarate, tert-butylhydroquinone, rosmarinic acid, and mitoquinone inhibited calcification through the attenuation of excess ROS production in VSMCs under osteogenic conditions, whereas NQO1 upregulation played a critical role in hydrogen-sulfide-mediated inhibition [38,39,40,41].

Here we showed that heme largely amplifies ROS production, and only slightly increases NQO1 expression, therefore we assumed that some other mechanism was implicated in heme-mediated inhibition of calcification in HuLECs. Heme is a strong inducer of HO-1, and we showed that inhibition of HO-1 activity with ZnPP or SnPP abolishes the anti-calcification effect of heme (Figure 6). HO-1-mediated heme degradation yields equimolar amounts of biliverdin, iron, and CO. Biliverdin is promptly converted to bilirubin. These heme degradation products have distinct biological effects, therefore we addressed whether these molecules influence OM-induced calcification in HuLECs. We showed that iron has a very strong, bilirubin has a mild, and CO has no inhibitory effect on OM-induced HuLECs calcification (Figure 7). This result is in agreement with the previously reported strong inhibitory effect of iron on osteogenic differentiation of VSMCs, osteoblasts, mesenchymal stem cells, and valve interstitial cells [27,28,42,43]. Iron metabolism in the lens and the role of iron in cataractogenesis are not thoroughly investigated and understood [44]. Nevertheless, the Lens Opacities Case-Control Study evaluated risk factors for age-related nuclear, cortical, posterior subcapsular, and mixed cataracts in 1380 participants and found that dietary iron intake decreased the risk of cortical, nuclear, and mixed cataracts [45].

A large number of studies have described the involvement of the Nrf2 antioxidant system in the prevention or attenuation of age-related diseases including atherosclerosis, vascular calcification, cataract formation, and macular degeneration [21,25,33,46,47,48]. There are numerous clinical trials targeting the Nrf2 system in diverse clinical conditions including cancer, chronic kidney disease, diabetes, aging problems, etc. [49].

Here we reported that heme-mediated activation of the Nrf2/HO-1 system inhibits HuLECs calcification, a mechanism that might contribute to cataract-associated lens calcification. Our result revealed that intracellular iron, released from heme during HO-1-catalyzed heme degradation, is the ultimate anti-calcification mediator.

Further investigations are needed for a better understanding of the calcification phenomenon of LECs. In particular, we completely lack information about the pyrophosphate (PPi) metabolism of LECs. Extracellular PPi is a very potent endogenous inhibitor of soft tissue calcification [50]. PPi is cleaved by ALP therefore high ALP expression is usually associated with low PPi level and calcification. We found elevated levels of ALP in calcifying HuLECs, therefore further experiments are needed to measure PPi levels and the involvement of this pathway in LECs calcification. Furthermore, we need to establish ex vivo organ culture and in vivo animal models of lens calcification. These experimental tools would be essential for further development of this field and could be extremely useful in testing therapeutic agents to prevent or attenuate lens calcification.

## 4. Materials and Methods

### 4.1. Materials

All the reagents were purchased from Sigma-Aldrich Co. (St. Louis, MO, USA) unless otherwise specified.

### 4.2. Cell Culture and Treatments

Immortalized human lens epithelial cells (HuLECs) were purchased from ATCC (Manassas, VA, USA). Cells were maintained in DMEM supplemented with 10% FBS, penicillin, and streptomycin (ScienceCell Research Laboratories, Carlsbad, CA, USA). Cells were maintained at 37 °C in a humidified atmosphere containing 5% CO_2_. Cells were grown to confluence and used from passages 3 to 4. Treatments were carried out in DMEM supplemented with 2% FBS (Ctrl). Osteogenic medium (OM) contained DMEM supplemented with 2% FBS, inorganic phosphate (Pi) (NaH2PO4-Na2HPO4, 2.5 mmol/L, pH 7.4) and Ca (0.3 mmol/L) in a form of CaCl_2_. Heme (H9039, Sigma) was dissolved in 20 mmol/L NaOH. Iron was introduced as ammonium ferric citrate (F5879, Sigma), dissolved in deionized water. Bilirubin (B4126, Sigma) was dissolved in NaOH. The Nrf2 inhibitor ML385 (SML1833, Sigma) was dissolved in DMSO. Tin protoporphyrin IX (SnPP, 16375, Cayman chemical, Ann Arbor, MI, USA) and zinc protoporphyrin IX (ZnPP, 691550-M, EMD Millipore Corp., Bedford, MA, USA) was dissolved in DMSO. The final concentration of NaOH was kept below 2 mmol/L and DMSO was less than 1% in all experiments. To deliver CO, we used tricarbonyl-dichloro-ruthenium (II) dimer also known as CO releasing molecule 2 (CORM2), [Ru_2_Cl_4_(CO)_6_] (288144, Sigma). CORM2 was dissolved in DMSO immediately before use, and it was administered every 12 h.

### 4.3. Alizarin Red (AR) Staining and Quantification

After washing with DPBS, the cells were fixed in 4% paraformaldehyde (16005, Sigma) and rinsed with deionized water thoroughly. Cells were stained with Alizarin Red S (A5533, Sigma) solution (2%, pH 4.2) for 20 min at room temperature. Excessive dye was removed by several washes in deionized water. To quantify AR staining in 96-well plates, we added 100 μL of hexadecyl-pyridinium chloride (C9002, Sigma) solution (100 mmol/L) to the wells and measured optical density (OD) at 560 nm using hexadecyl-pyidinium chloride solution as blank.

### 4.4. Quantification of Ca Deposition

Cells grown on 96-well plates were washed twice with DPBS and decalcified with HCl (30721, Sigma, 0.6 mol/L) for 30 min at room temperature. Ca content of the HCl supernatants was determined by QuantiChrome Calcium Assay Kit (DICA-500, Gentaur, Kampenhout, Belgium). Following decalcification, cells were washed twice with DPBS and solubilized with a solution of NaOH (S8045, Sigma, 0.1 mol/L) and sodium dodecyl sulfate (11667289001, Sigma, 0.1%), and the protein content of samples was measured with BCA protein assay kit (23225, Pierce Biotechnology, Rockford, IL, USA). Ca content of the cells was normalized to protein content and expressed as mg/mg protein. The observer who performed all the Ca measurements was blinded to the group assignment.

### 4.5. Quantification of OCN

For OCN detection, the ECM of the cells grown on 6-well plates was dissolved in 100 μL of EDTA (E6758, Sigma, 0.5 mol/L, pH 6.9). OCN content of the EDTA-solubilized ECM samples was quantified by an enzyme-linked immunosorbent assay (ELISA) (DY1419-05, DuoSet ELISA, R&D, Minneapolis, MN, USA) according to the manufacturer’s protocol. The observer who performed all the ELISA measurements was blinded to the group assignment.

### 4.6. Western Blot Analysis

For evaluation of ALP, Nrf2, HO-1, NQO1, and RUNX2 protein expressions, cells were lysed in Laemmli lysis buffer (38733, Sigma). Whole-cell lysates were electrophoresed in 10% SDS-PAGE, then blotted onto a nitrocellulose membrane (1060003, Amersham Proton, GE Healthcare, Chicago, IL, USA). Western blotting was performed with the use of an anti-ALP antibody (sc30203, Santa Cruz Biotechnology, Inc., Dallas, TX, USA) at a 1:1000 dilution, anti-HO-1 antibody (70081, Cell Signaling Technology, Leiden, The Netherlands) at a 1:1000 dilution, anti-Nrf2 antibody (16396-1-AP, Proteintech, Rosemont, IL, USA) at a 1:1000 dilution, anti-NQO1 antibody (#3187, Cell Signaling Technology, Leiden, The Netherlands), at a 1:1000 dilution, anti-RUNX2 antibody (GTX81326, GeneTex International Corporation, Irvine, CA, USA) at a 1:1000 dilution followed by HRP-labeled anti-rabbit or anti-mouse IgG secondary antibodies (NA-934 and NA-931, Amersham Biosciences Corp., Piscataway, NJ, USA). After binding of the primary antibodies, membranes were incubated with horseradish peroxidase-linked rabbit (NA-934) and mouse IgG (NA-931) (Amersham, GE Healthcare) at 0.5 µg/mL concentration. Antigen–antibody complexes were visualized with enhanced chemiluminescence system Clarity Western ECL (170-5060, BioRad, Hercules, CA, USA). Chemiluminescent signals were detected conventionally on an X-ray film or digitally by using a C-Digit Blot Scanner (LI-COR Biosciences, Lincoln, NE, USA). After detection, the membranes were stripped and re-probed for β-actin using an anti-β-actin antibody (sc-47778, Santa Cruz Biotechnology Inc., Dallas, TX, USA) at 0.2 µg/mL concentration. Western blots were repeated three times with independent sample sets and blots were quantified by using the inbuilt software on the C-Digit Blot Scanner.

### 4.7. Intracellular ROS Measurement

ROS production was monitored by using the 5-(and-6)-chloromethyl-2′,7′-dichlorodihydro-fluorescein di-acetate, acetyl ester (CM-H2DCFDA) assay (C6827, Life Technologies, Carlsbad, CA, USA) as previously described. After a 4-h pre-treatment, cells were washed with DPBS and loaded with CM-H2DCFDA (10 µmol/L, 30 min, in the dark). Cells were washed thoroughly with DPBS and fluorescence intensity was measured every 30 min for 3 h applying 488 nm excitation and 533 nm emission wavelengths. In some experiments, we applied the ROS inhibitor N-acetyl cysteine (NAC, 3 mmol/L), (A9165, Sigma).

### 4.8. Statistical Analysis

Group size was equal in all experiments and no data points were excluded from the analysis. Data are presented as mean ± SD with individual data points. Statistical analyses were performed with GraphPad Prism software (version 8.01, San Diego, CA, USA). Comparisons between more than two groups were carried out by one-way ANOVA followed by Tukey’s multiple-comparisons test. Time-course experiments were analyzed by two-way ANOVA followed by Tukey’s multiple-comparisons test. A value of *p* < 0.05 was considered significant.

## 5. Conclusions

Hydroxyapatite, the major inorganic constituent of bone tissue, is present in calcified cataractous lenses. Hydroxyapatite can form upon osteogenic differentiation of HuLECs. The conclusion of the current work is that upregulation of the Nrf2/HO-1 antioxidant system inhibits high phosphate-induced osteogenic differentiation and calcification of HuLECs. Further studies are needed to investigate whether the Nrf2/HO-1 axis could serve as a therapeutic target in the prevention of lens calcification in vivo.

## Figures and Tables

**Figure 1 pharmaceuticals-15-00493-f001:**
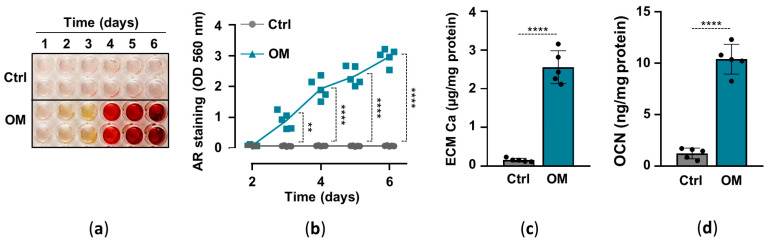
(**a**–**d**) Confluent HuLECs (passage 5–8) were maintained in control (Ctrl) or osteogenic conditions (OM). (**a**,**b**) Time course of Ca deposition evaluated by AR staining. Representative staining and quantification from 4 independent experiments are shown. (**c**) Ca content of the HCl-solubilized ECM (day6). (**d**) OCN level in EDTA-solubilized ECM (day6). (**c**,**d**) Data are expressed as mean ± SD, from 5 independent experiments. (**b**) Multiple *t*-tests to compare Ctrl and OM at each time point were performed to obtain *p* values. (**c**,**d**) *t*-test was used to obtain *p* values. ** *p* < 0.01, **** *p* < 0.001.

**Figure 2 pharmaceuticals-15-00493-f002:**
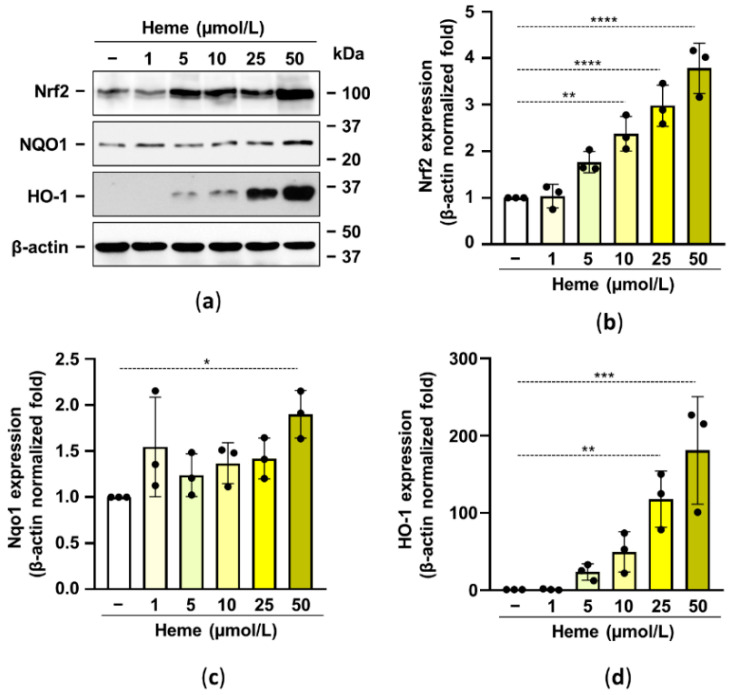
Heme induces the Nrf2/HO-1 axis in HuLECs. (**a**–**d**) Confluent HuLECs (passage 5–8) were treated with heme (0–50 μmol/L). Protein expressions of Nrf2, NQO1, HO-1, and β-actin were determined from whole-cell lysates (12 h). (**a**) Representative Western blots out of three. (**b**–**d**) β-actin-normalized relative expression of Nrf2, NQO1, and HO-1. Bars represent mean ± SD. Ordinary one-way ANOVA followed by Dunett’s multiple comparison test was used to obtain *p* values. * *p* < 0.05, ** *p* < 0.01, *** *p* < 0.005, **** *p* < 0.001.

**Figure 3 pharmaceuticals-15-00493-f003:**
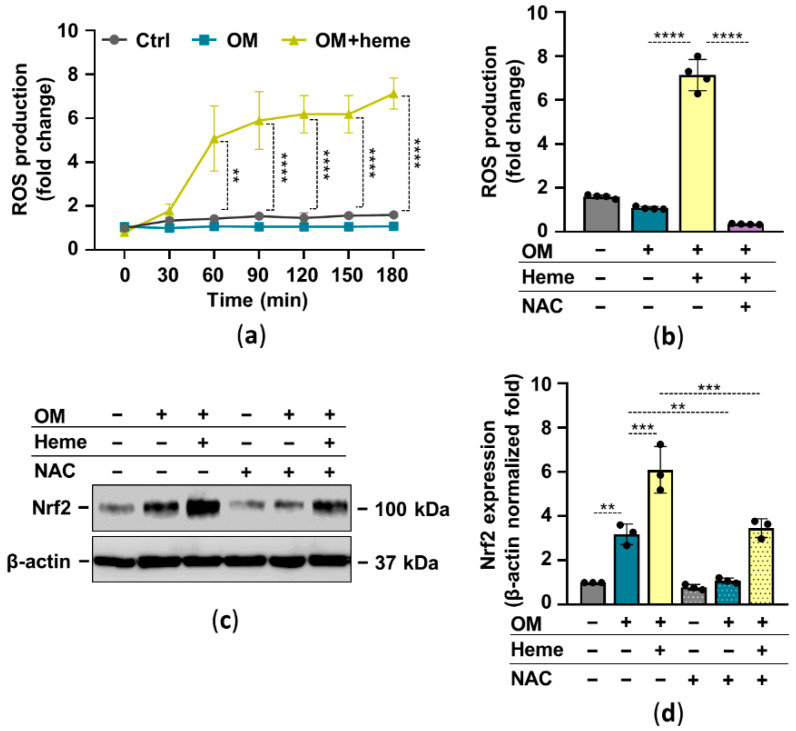
Heme-mediated ROS production plays a role in Nrf2 activation. Confluent HuLECs (passage number 5–8) were treated with OM and OM + heme (25 µmol/L) in the presence or absence of NAC (5 mmol/L). (**a**) Kinetics of intracellular ROS production was monitored for 3 h. (**a**,**b**) Data points are derived from 4 independent experiments. (**b**) ROS production (3 h), *n* = 3. (**c**,**d**) Protein expressions of Nrf2 and β-actin in whole-cell lysates (12 h). (**c**) Representative Western blots (*n* = 3). (**d**) β-actin-normalized Nrf2 expression. Data are expressed as mean ± SD. Ordinary one-way ANOVA followed by Tukey’s multiple comparison test was used to obtain *p* values. ** *p* < 0.01, *** *p* < 0.005, **** *p* < 0.001.

**Figure 4 pharmaceuticals-15-00493-f004:**
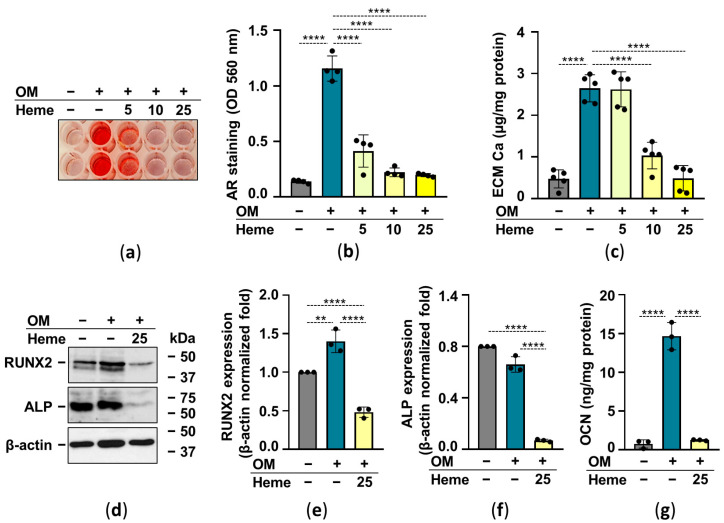
Heme inhibits OM-induced ECM calcification of HuLECs. (**a**–**g**) Confluent HuLECs (passage number 5–8) were treated with Ctrl, OM, and OM + heme as indicated. (**a**,**b**) Dose-dependent effect of heme (5, 10, 25 µmol/L) on OM-induced Ca deposition in the ECM evaluated by AR staining (day6). Representative images and quantification from 4 independent experiments are shown. (**c**) Ca content of the HCl-solubilized ECM (day6). Data points represent independent experiments. (**d**–**f**) Protein expressions of Runx2 and ALP were determined from whole-cell lysates (day6). Membranes were re-probed for β-actin. (**d**) Representative Western blots (*n* = 3). (**e**,**f**) Relative expressions of Runx2 and ALP normalized to β-actin are shown. (**g**) OCN level in EDTA-solubilized ECM (day6) from 3 independent experiments. Data are expressed as mean ± SD. Ordinary one-way ANOVA followed by Tukey’s multiple comparison test was used to obtain *p* values. ** *p* < 0.01, **** *p* < 0.001.

**Figure 5 pharmaceuticals-15-00493-f005:**
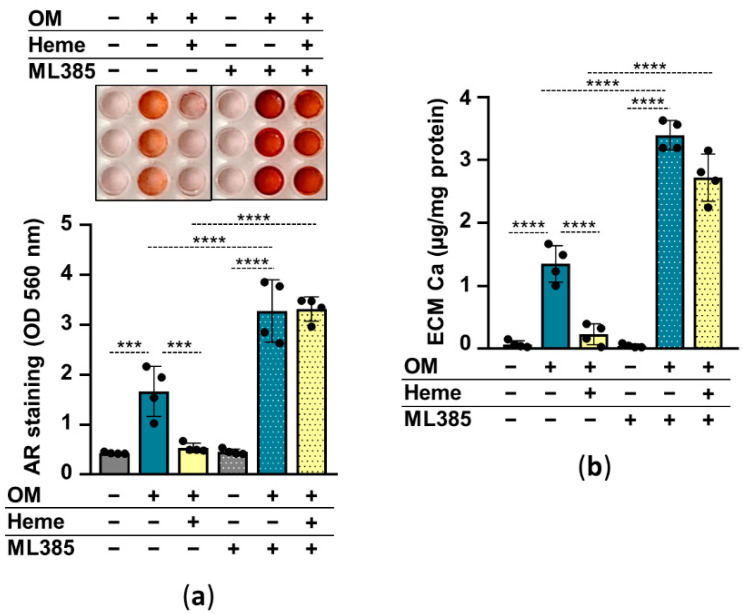
Inhibition of Nrf2 diminishes the anti-calcification effect of heme. (**a**,**b**) Confluent HuLECs (passages 5–8) were pretreated with vehicle or ML385 (10 µmol/L) for 3 h. After the pretreatments cells were maintained in Ctrl, OM, or OM + heme (10 µmol/L) conditions. (**a**) Representative AR staining and quantification from 4 independent experiments are shown (day5). (**b**) Ca content of the HCl-solubilized ECM is presented (day5). Data points are derived from 4 independent experiments. Data are expressed as mean ± SD. Ordinary one-way ANOVA followed by Tukey’s multiple comparison test was used to obtain *p* values. *** *p* < 0.005, **** *p* < 0.001.

**Figure 6 pharmaceuticals-15-00493-f006:**
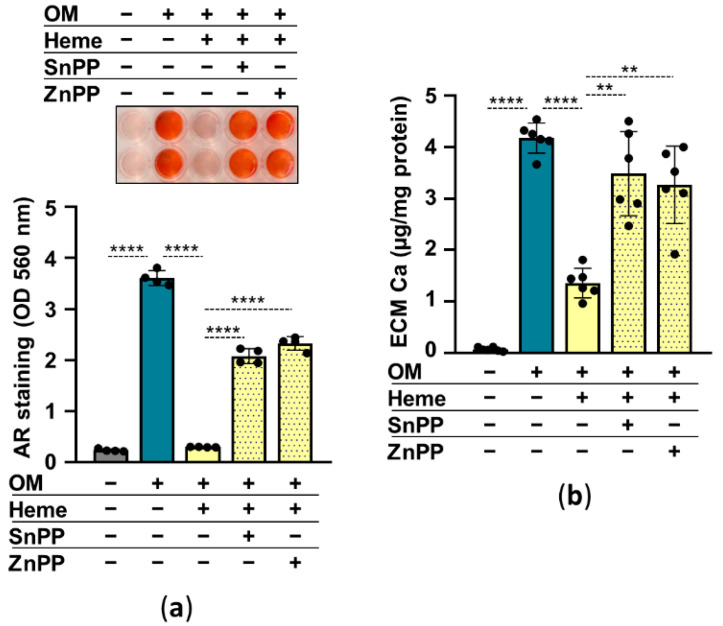
Inhibition of HO-1 enzyme activity diminishes the anti-calcification effect of heme. (**a**,**b**) Confluent HuLECs (passages 5–8) were pretreated with vehicle or SnPP (10 µmol/L) or ZnPP (10 µmol/L) for 3 h. After the pretreatments cells were maintained in Ctrl, OM, or OM + heme (10 µmol/L) conditions. (**a**) Representative AR staining and quantification (day5). (**b**) Ca content of the HCl-solubilized ECM is presented (day5). Data points are derived from 4 independent experiments. Data are expressed as mean ± SD. Ordinary one-way ANOVA followed by Tukey’s multiple comparison test was used to obtain *p* values. ** *p* < 0.01, **** *p* < 0.001.

**Figure 7 pharmaceuticals-15-00493-f007:**
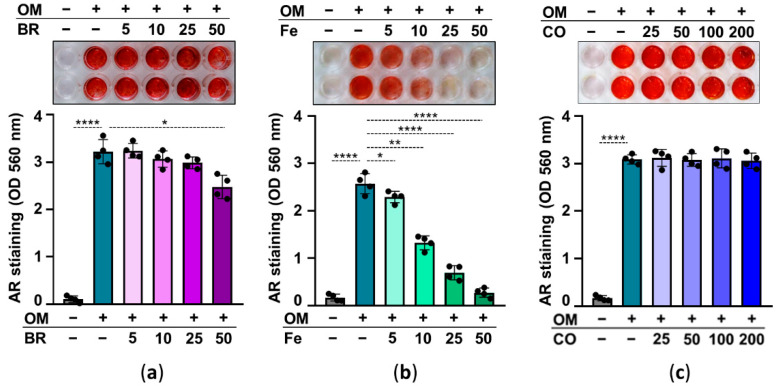
Effect of heme degradation products on OM-induced HuLECs calcification. (**a**–**c**) Confluent HuLECs (passage 5–8) were treated with OM in the presence of bilirubin (BR; 5–50 µmol/L), iron (FeSO_4_; 5–50 µmol/L), or CO (CORM; 25–200 µmol/L). Calcification was assessed by AR staining (day5). Representative images from four independent experiments and quantification are shown. Data are expressed as mean ± SD. Ordinary one-way ANOVA followed by Tukey’s multiple comparison test was used to obtain *p* values. * *p* < 0.05, ** *p* < 0.01, **** *p* < 0.001.

## Data Availability

All data generated or analysed during this study are included in this published article.

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
