# Peer review of "Activation of Nrf2/HO-1 Antioxidant Pathway by Heme Attenuates Calcification of Human Lens Epithelial Cells"

_pharmaceuticals, 2022, doi:10.3390/ph15050493_

Round 1

Reviewer 1 Report

Here Chowdhury and colleagues performed a comprehensive and insightful study deciphering the mechanisms of heme-mediated inhibition of osteogenic differentiation. Using a model of human lens endothelial cells, authors showed their mineralisation by Ca/P supersaturated medium and abrogation of the osteogenic differentiation by adding heme (mainly because of iron activity), further pinpointing the mechanisms of protective heme action (Fenton-reaction derives production of reactive oxygen species provoking Nrf2 activation with the downstream NAD(P)H quinone dehydrogenase 1 and heme oxygenase-1 upregulation). 

The results seem solid and robust, as authors applied an impressive armamentarium of specific techniques (alizarin red S staining, quantification of calcium content, Western blotting profiling using specific antibodies with the further semi-quantitative analysis, ELISA, and intracellular ROS measurement). Statistical analysis also looks sound and all experiments are in concordance. Figures are drawn concisely and well illustrate study findings. I cannot see any glaring omissions in methodology or data presentation as it is indeed comprehensive and reliable.

Authors conclude that intracellular iron and iron released from heme during HO-1 catalysed heme degradation is an efficient natural anti-calcification mediator; this can be potentially used to prevent osteogenic differentiation in extraskeletal tissues. Potential therapeutic applications might be added to the Discussion. 

The key role of iron in activating Nrf2 can also be of crucial importance for endothelial physiology and vascular biology, as Nrf2 is a major atheroprotective transcription factor and its physiological modulation can be harnessed for the prevention and reversal of atherosclerosis. Can we govern iron-mediated Nrf2 expression through the respective diet modulations or pharmacological interventions? If yes, this promotion of Nrf2 might protect against both atherosclerosis and extraskeletal calcification which represent 2 vascular pathologies mainly responsible for deteriorated quality of life and cardiovascular death.

To summarise, I suppose that the findings obtained by the authors can be employed by far more than in ocular biology and suggest to significantly extend the discussion given the abovementioned applications of the authors' results. Authors are also encouraged to talk about the physiological interactions between the circulating heme/iron and endothelial cells. The paper can be accepted upon these revisions and is of particular importance. I would recommend this paper for the further promotion by the journal as it sheds light on the important and previously neglected anti-calcification and anti-atherosclerotic mechanism in a clear and concise manner.

It would be also good to add the Conclusions section highlighting further directions in interventional upregulation of Nrf2 and possibly indicating further plans of the authors' research group in this field.

Author Response

Response to reviewer

We thank to the reviewer for the positive review and the insightful suggestions.

As suggested by the reviewer we extended the discussion and addressed the important topics.

The major changes have been made on the discussion are the following:

“A large number of studies have described the involvement of the Nrf2 antioxidant system in the prevention or attenuation of age-related diseases including atherosclerosis, vascular calcification, cataract formation and macular degeneration [21,25,33,47–49]. There are numerous clinical trials targeting the Nrf2 system in diverse clinical conditions including cancer, chronic kidney disease, diabetes, aging problems, etc [50].” 

“Here we reported that heme-mediated activation of the Nrf2/HO-1 system inhibits HuLECs calcification, a mechanism that might contribute to cataract-associated lens calcification. Our result revealed that intracellular iron, released from heme during HO-1-catalyzed heme degradation, is the ultimate anti-calcification mediator.”

“Further investigations are needed for better understanding the calcification phenomenon of LECs. In particularly, we completely lack information about pyrophosphate (PPi) metabolism of LECs. Extracellular PPi is a very potent endogenous inhibitor of soft tissue calcification [51]. PPi is cleaved by ALP therefore high ALP expression is usually associated with low PPi level and calcification. We found elevated level of ALP in calcifying HuLECs, therefore further experiments are needed to measure PPi levels and the involvement of this pathway in LECs calcification. Furthermore, we need to establish ex vivo organ culture and in vivo animal models of lens calcification. These experimental tools would be essential for further development of this field, and could be extremely useful in testing therapeutic agents to prevent or attenuate lens calcification.”  

We hope that you will find our revised manuscript suitable for publishing in Pharmaceuticals. Thank you again for your time and effort to review our manuscript.

Best regards,

Viktória Jeney

MTA-DE Lendület Vascular Pathophysiology Research Group

Research Centre for Molecular Medicine

Faculty of Medicine, University of Debrecen

Debrecen, Hungary

Reviewer 2 Report

 This paper by Chowdhury et al, describes the role of the oxidative stress and the Nrf2/HO-1 pathway in the regulation of ECM calcification associated with osteogenic differentiation of cultured human lens epithelial cells.

This paper is a simple and straight forward investigation with a classic approach with induction and inhibition using a cell culture model that provide some molecular information on a human pathology (i.e. ectopic calcification associated with cataract). This is also an incremental step following a previous papers showing osteogenic differentiation of huLECs.

Main comments.

  • Since the pathway investigated is not the only one regulating ectopic calcification and because the authors have shown in this paper and also in Balogh et al, 2016 that alkaline phosphatase is dysregulated in their model, it would be very informative to investigate whether the ABCC6/ENPP1/CD73 – TNAP axis is also involved which includes measuring pyrophosphate concentration in the culture medium (see PMID: 24277820, 30155470, 32042814and 30155470 as initial references).
  • How was chosen the concentration of heme and degradation compound to use in cell culture? Are these physiological? Please indicate. 
  • For many of the figures it’s unclear if the sample replicates (Most experiment show 4 data points) are 4 samples of the same culture or from 4 individual cultures ? This should be clarified in the methods and legends.

 Minor comments:

Lane 49: cataract should not be considered as a type of calcification but rather calcification is a pathological manifestation associated with cataract. Indeed, calcification is present as secondary manifestations in many other pathologies.  

Lane 51-52: calcification in the vasculature (and VSMC) has received a lot of attention but so does mineralization in heritable diseases which has provided new and interesting information on the subject. This sentence should be revised

The use of punctuation throughout the manuscript is sub-par and should be worked on.

Author Response

Response to reviewer

We thank to the reviewer for the positive review and the helpful suggestions. We would like to take the opportunity to respond point by point to the reviewer’s specific comments.

Comments and answers:

  1. Since the pathway investigated is not the only one regulating ectopic calcification and because the authors have shown in this paper and also in Balogh et al, 2016 that alkaline phosphatase is dysregulated in their model, it would be very informative to investigate whether the ABCC6/ENPP1/CD73 – TNAP axis is also involved which includes measuring pyrophosphate concentration in the culture medium (see PMID: 24277820, 30155470, 32042814and 30155470 as initial references).

Answer: Thank you for the comment. We searched the literature and could not find any information about pyrophosphate metabolism in lens epithelial cells. We believe that not only pyrophosphate level, but the whole pathway is worth exploring in the future in a separate study. We discuss this idea in a newly given paragraph of future directions in the revised manuscript:

“Further investigations are needed for better understanding the calcification phenomenon of LECs. In particularly, we completely lack information about pyrophosphate (PPi) metabolism of LECs. Extracellular PPi is a very potent endogenous inhibitor of soft tissue calcification [51]. PPi is cleaved by ALP therefore high ALP expression is usually associated with low PPi level and calcification. We found elevated level of ALP in calcifying HuLECs, therefore further experiments are needed to measure PPi levels and the involvement of this pathway in LECs calcification.”     

  1. How was chosen the concentration of heme and degradation compound to use in cell culture? Are these physiological? Please indicate. 

Answer: Heme was used to activate the Nrf2/HO-1 system, and the concentration of heme was chosen based on previous publications. Although heme is a ubiquitous molecule, and its concentration can be quite high in cells, but that heme is protein-bound. Therefore the heme concentration used in this work is not physiological, rather pathological.  

Heme degradation yields equimolar amounts of biliverdin, iron and CO, and the biliverdin is converted into equimolar amount of bilirubin. Therefore we used iron and bilirubin at the same concentrations as heme. As for CO, we used the CO donor molecule CORM that releases CO slowly, therefore we increased the concentration of this compound.  

We indicated this in the result section:

“Based on previous publications we used heme at a supraphysiological concentration of 1-50 µmol/L to activate the Nrf2/HO-1 antioxidant pathway.”   

  1. For many of the figures it’s unclear if the sample replicates (Most experiment show 4 data points) are 4 samples of the same culture or from 4 individual cultures? This should be clarified in the methods and legends.

Answer: Thank you for the comment. We show sample replicates on alizarin red quantification panels, because the staining intensity varies a lot experiment by experiment. But these experiments were repeated at least 4 times with the same trend of staining. On the other panels one point represent the result of an independent experiments with 3-6 technical replicates. We clarified this issue in the figure legends.  

  1. Lane 49: cataract should not be considered as a type of calcification but rather calcification is a pathological manifestation associated with cataract. Indeed, calcification is present as secondary manifestations in many other pathologies.  

Answer: Thank you for the comment, we deleted the indicated sentence, and rephrased the paragraph (see below). 

  1. Lane 51-52: calcification in the vasculature (and VSMC) has received a lot of attention but so does mineralization in heritable diseases which has provided new and interesting information on the subject. This sentence should be revised.

Answer: Thank you for the comment. We thoroughly revised this part of the introduction and we inserted a paragraph addressing heritable diseases with ectopic calcification:

“Deposition of hydroxyapatite in soft tissues or so called ectopic calcification is a pathologic event which can occur in both genetic and acquired clinical conditions. For long time ectopic calcification was considered to be a passive process but recent discoveries proved that it is not the case, and identified numerous regulators of the calcification process. For example studies on patients with rare inherited calcification syndromes such as pseudoxanthoma elasticum, generalized arterial calcification of infancy, and arterial calcification due to deficiency of CD73 led to the identification of inorganic pyrophosphate (PPi) as the principal physiologic inhibitor of hydroxyapatite deposition in soft tissues [8–11]. These hereditary disorders are caused by monogenic mutations in genes encoding key proteins in PPi metabolism such as ATP binding cassette subfamily C member 6, ectonucleotide pyrophosphatase/phosphodiesterase 1 and CD73 proteins, leading to reduced circulating levels of PPi [8–11].

Besides the genetic origin, ectopic calcification, especially vascular calcification is associated with chronic diseases such as renal failure and diabetes [12]. Additionally, aging is also an important predisposing factor for vascular calcification [13].

  1. The use of punctuation throughout the manuscript is sub-par and should be worked on.

Answer: Thank you for the notice, we have worked on it. 

We hope that you will find our revised manuscript suitable for publishing in Pharmaceuticals. Thank you again for your time and effort to review our manuscript.

Best regards,

Viktória Jeney

MTA-DE Lendület Vascular Pathophysiology Research Group

Research Centre for Molecular Medicine

Faculty of Medicine, University of Debrecen

Debrecen, Hungary
